# Self-sharpening induces jet-like structure in seafloor gravity currents

R.M. Dorrell [1], J. Peakall[2], S.E. Darby[3], D.R. Parsons[1], J. Johnson[1], E.J. Sumner[4], R.B. Wynn[5], E. Özsoy[6,7] & D. Tezcan[6]

Gravity currents are the primary means by which sediments, solutes and heat are transported across the ocean-floor. Existing theory of gravity current flow employs a statistically-stable model of turbulent diffusion that has been extant since the 1960s. Here we present the first set of detailed spatial data from a gravity current over a rough seafloor that demonstrate that this existing paradigm is not universal. Specifically, in contrast to predictions from turbulent diffusion theory, self-sharpened velocity and concentration profiles and a stable barrier to mixing are observed. Our new observations are explained by statistically-unstable mixing and self-sharpening, by boundary-induced internal gravity waves; as predicted by recent advances in fluid dynamics. Self-sharpening helps explain phenomena such as ultra-long runout of gravity currents and restricted growth of bedforms, and highlights increased geohazard risk to marine infrastructure. These processes likely have broader application, for example to wave-turbulence interaction, and mixing processes in environmental flows.

[1] Energy and Environment Institute, University of Hull, Hull HU6 7RX, UK. [2] School of Earth and Environment, University of Leeds, Leeds LS2 9JT, UK. [3] Geography and Environment, University of Southampton, Southampton SO17 1BJ, UK. [4] Ocean and Earth Science, University of Southampton, Southampton SO14 3ZH, UK. [5] National Oceanography Centre, Southampton SO14 3ZH, UK. [6] Institute of Marine Sciences, Middle East Technical University, Mersin 33731 Erdemli, Turkey. [7] Eurasia Institute of Earth Sciences, İstanbul Technical University, İstanbul 80626, Turkey. Correspondence and requests for materials should be addressed to R.M.D. (email: r.dorrell@hull.ac.uk)

Seafloor gravity currents are a key geophysical flow critical for transporting sediment, salinity, heat, organic carbon, oxygen, nutrients and pollutants within the world's oceans[1–6]. These flows are driven by density differences arising from variations in suspended sediment concentration, salinity and/or temperature[1,2]. As some of the largest flows on the Earth's surface they are fundamental to surface process dynamics and have been studied extensively using laboratory[7–13] and numerical[14–19] models. A fundamental constraint on understanding these flows is their inaccessible seafloor location; the lack of field data means that the applicability of scaled laboratory and numerical models to real-world gravity currents remains largely unknown[6,20]. However, recent technological advances in autonomous underwater vehicles (AUVs) now affords the opportunity to acquire uniquely detailed field scale measurements of gravity currents[21]. Advances made by such studies[22] are essential for validating flow models and predicting the impact of gravity currents on natural environments.

Gravity current dynamics depend on variations in the excess density of the flow, i.e. stratification, and shear and turbulent mixing resulting from the flow's interactions with the seafloor and the ambient seawater. Gravity currents are therefore split into two regimes, a lower shear layer, forced by interactions with the seafloor, and an upper shear layer dependent on flow stratification and interactions with the ambient fluid (Fig. 1a). The stationary seafloor and the decrease of excess density with height above the seafloor imply zero velocity boundary conditions at both the lower and upper limits of the flow, thus a two-dimensional flow is often assumed[1,17–19] (the first of two common key assumptions on gravity current dynamics). The zero-shear, velocity maximum between these two regimes defines the lower and upper shear layer boundary. The upper shear layer has been studied in detail; with laboratory studies suggesting that the flow velocity is well approximated as a free-jet, with an exponentially decreasing (concave up) profile, with a Gaussian decay with distance from the velocity maximum, in subcritical[9,11,18] and

supercritical flows[11]; albeit it has been argued that linear or exponential decay may occur in some supercritical currents[18]. In comparison, in the lower shear layer there is a lack of empirical data to support development of theoretical models of gravity current flow dynamics. Although the relative size of the upper and lower shear layers varies with environment and flow conditions[11], the velocity maximum is often located close to the bed[5], constraining resolution of flow dynamics in the lower shear layer[9–13]. Therefore, the lower shear layer flow has been approximated by a boundary-layer flow with an inner and outer region, analogous to studies of open-channel flow (Fig. 1a). The flow in the inner region is characterised by a viscous sub-layer to turbulent flow transition, assumed to follow the law of the wall[9]. In the outer region the flow follows the concave up profile of the inner region; a result of short range (in comparison to the length scale over which velocity varies) isotropic turbulent fluctuations generating a down-velocity-gradient momentum flux[23,24] (i.e. towards the bed in the lower shear layer and towards the flow ambient fluid interface in the upper shear layer). Where models do not fully resolve turbulent fluid motion the frictional turbulent diffusion of momentum, and analogously diffusion of material transported by the flow, are parameterized by a positive[23] eddy diffusivity model (the second of two common key assumptions on gravity current flow dynamics).

In contrast, the dynamics of another prevalent type of geophysical flow, zonal jets (Fig. 1b), are driven by up-velocity-gradient momentum transport (i.e. towards the jet core) as a result of anti-frictional radiation stresses, a process that has previously been ascribed as "negative viscosity"[25,26]. Only within the last decade has the previously enigmatic occurrence of permanent, planetary-scale zonal jets been fully explained by wave-turbulence interaction[27]. Planetary Rossby waves, and other types of dispersive waves[26], where phase speed varies with wavelength, can generate systematic correlations of turbulence (radiation stresses) and enable up-gradient momentum transport[28]. Dispersive waves propagate on gradients of potential vorticity (PV);

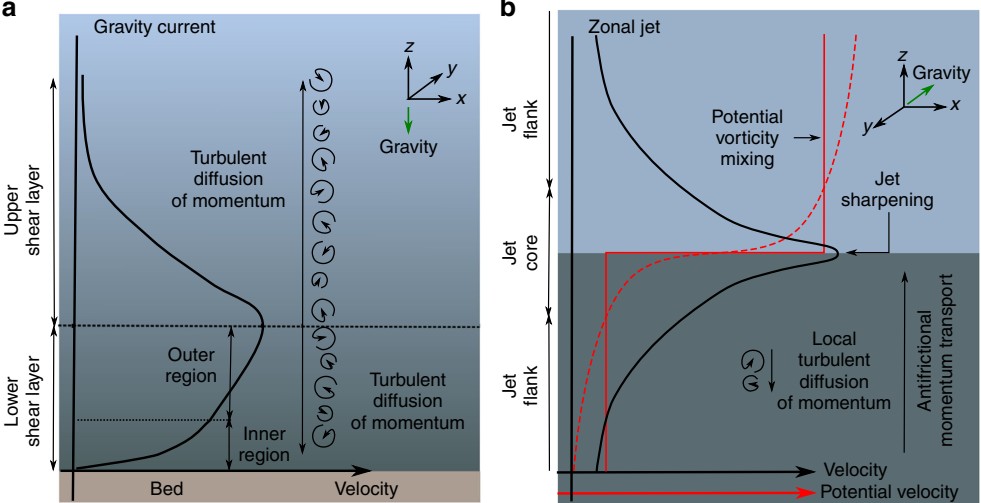

**Fig. 1** Schematic velocity profiles of two different geophysical flows. A standard model for seafloor gravity currents (**a**), where turbulent fluctuations diffuse momentum down the vertical gradient of primary flow velocity, resulting in a concave upwards velocity profile (solid black line) in the inner and outer regions of the lower shear layer. The size of the lower and upper shear layers is not drawn to scale, varying with flow and environment conditions[11]. Depicted by background shading, turbulent mixing causes the scalar quantities transported by the flow (e.g., heat, solutes or particulates) to take on a smooth gradient. In contrast, in an idealised zonal jet (**b**), the self-organisation of turbulence, by dispersive waves, results in up-velocity-gradient momentum transport whilst irreversible wave breaking transfers wave momentum into the mean flow[26]. Breaking dispersive waves drive homogenisation of the potential vorticity profile, reinforcing flow sharpening, as denoted by red dashed to solid line. Strong gradients in potential vorticity, mixed by dispersive wave breaking act as an eddy transport barrier, preventing the mixing of the scalar quantities transported by the flow, resulting in a strongly stratified flow (depicted by background shading)

PV being a measure of circulation in a stratified fluid (including planetary rotation) that is conserved in the absence of frictional dissipation (computed by the product of density stratification and absolute vorticity). Irreversible breaking of Rossby, or other dispersive waves, near a critical layer where the background flow speed tends to the wave phase speed[29] result in deposition of wave momentum, with concomitant changes in angular momentum distribution[26], and generation of mean flow[30,31]. Wave breaking homogenises PV and encourages further mixing[32]. Consequently, at the boundaries of these mixing layers PV gradients are intensified[33] (Fig. 1b). Strong PV gradients provide a dispersive wave restoring mechanism, i.e. Rossby-wave elasticity[26]. Self-organisation of zonal jets is thus inbuilt; if the PV profile is disturbed then shear induced wave breaking on the jet flanks remixes PV inhomogeneity, providing a feedback mechanism to re-sharpen and narrow the jet core[34,35] (cf. the Special Collection in Jets and Annular Structures in Geophysical Fluids[26]). Furthermore, as eddy transport requires a PV anomaly larger than any PV inhomogeneity, strong PV gradients also act as an eddy transport barrier[26], inhibiting the mixing of momentum and any material (or scalar quantities) transported by the flow[32] (Fig. 1b).

Self-organisation of turbulent flows by dispersive waves is not limited to the formation and self-sharpening of zonal jets by Rossby waves, nor to flows in rotational frames of reference[26]. Prominent examples of self-organisation in (geo)-physical flows include the Quasi-Biennial Oscillation (QBO) of the equatorial zonal wind[36], the Phillips effect observed in oceans and the atmosphere[26,37] and drift wave-zonal flow (DW-ZF) in plasmas[38]. In the QBO gravity waves formed in the troposphere propagate into the stratosphere where they are absorbed at a critical layer. The upwards propagation of momentum, reinforcing zonal winds, results in the downwards migration of the critical layer. As the critical layer approaches the wave source, the wind direction is reversed[39]; the reversal of flow direction demonstrated by the celebrated laboratory experiment of Plumb and McEwan[40]. The Phillips effect describes how, under certain conditions, homogeneous mixing results in inhomogeneous stratification and stabilised density layering[41,42]. This density layering may be attributed to a positive feedback between weakening of the buoyancy gradient in a mixing layer and gravity wave elasticity, forming a barrier to transport across layer interfaces (in a similar fashion to eddy transport barriers in zonal jets[26]). In fusion plasmas, electrostatic turbulence and electron diamagnetic drift results in drift waves[38]. As in atmospheric jets, drift waves organise ordered motion like zonal flow (DW-ZF)[43] and associated plasma transport[38]. Indeed, it is noted that any systematic correlation of turbulent fluctuations can result in self-organisation[28,39], as long there is a process that causes irreversibility in the flow.

Here we present novel field data from an active seafloor gravity current suggesting that, at natural-scales, the dynamics of some stratified seafloor gravity currents are self-organised and strikingly similar to those of zonal jets. Our data reveal that the fundamental assumptions of diffusive mixing and quasi-two-dimensionality that underpin our present understanding of gravity current dynamics may be inappropriate.

## Results

**Field site.** Data were collected within a seafloor gravity current located at the exit of the Strait of Bosphorus, where high salinity Mediterranean water flows, via the Marmara Sea, into the comparatively lower salinity water of the south west Black Sea continental shelf, Fig. 2a. In comparison to existing studies of seafloor gravity currents[4,6], the quasi-permanent flow in the south west Black Sea presents a unique natural laboratory to study the dynamics of a field-scale seafloor gravity current in unprecedented detail. Moreover, data were obtained by advancing the state-of-the-art for deployment of remote monitoring technology. The gravity current is initially entrenched within a 15 km long single-thread channel[16] before flowing for at least a further 50 km through a shallow anastomosed channel network[44] (Fig. 2a). The flow data presented in this study were acquired within a 6 h time period, on the 5th July 2013, at a location ~35 km downstream from the Strait of Bosphorus where the channel floor is ornamented by high aspect-ratio, ~200:1, sedimentary bedforms (Fig. 2b). The seafloor over the bedform region dips downstream with a mean gradient of $1.2 \times 10^{-4}$. Multiple transects of flow

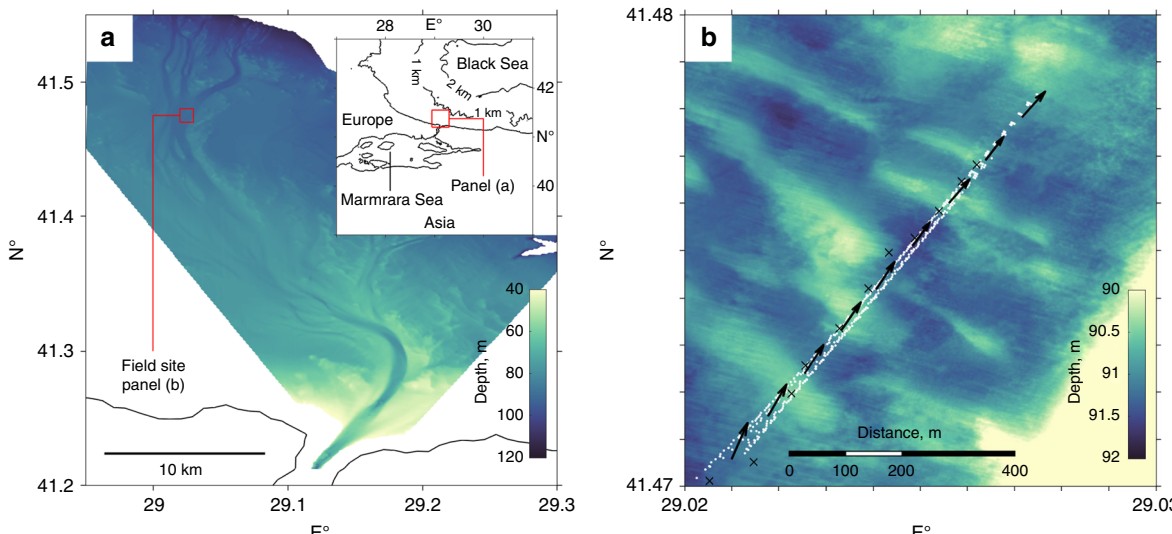

**Fig. 2** Field site location and bathymetric maps. Field site location in the channelized gravity current system, SW Black Sea shelf (**a**). Inset shows geographical location. The density-driven exchange flow, driven by salinity differences between the Mediterranean and Black Sea[22], has created a self-formed channelized network. Multi Beam Echo Sounder (MBES) bathymetry of the channel floor bedforms, (**b**), investigated here (see methods), where white dots denote the Autonomous Underwater Vehicle (AUV) tracks used in this study to obtain detailed flow velocity measurements, black arrows denote mean flow direction and black crosses denote conductivity-temperature-depth (CDT) cast locations

velocity data were measured using an Acoustic Doppler Current Profiler (ADCP) mounted on an AUV Autosub III[21], which was deployed from the R/V Pelagia. Within resolution mean flow was unchanging between transects; a time average map of velocity was compiled from >100,000 individual measurements derived from 16 repeat AUV transects (see methods, the AUV passing as close as 5 m above the seafloor in order to maximise data in the lower shear layer). Flow density was derived from 11 transect coincident conductivity-temperature-depth (CDT) casts (see methods) providing a two-dimensional profile of flow density. The high-resolution two-dimensional transects of velocity, and density, are a major advance on extant data sets, the latter being limited to low resolution or at a point data[4,6,16]. Since the velocity maximum was located at ~50% of the sampled flow depth, just under half of the flow velocity measurements are, for the first time in a field-scale gravity flow, located within the lower shear layer, providing an unprecedented dataset of flow dynamics below the velocity maximum. Although, the first ~1 m of the flow above the seafloor is lost due to acoustic side-lobe interference between the ADCP and bed (see methods) no evidence for flow separation in the lee of the bedforms was observed (Fig. 3a). As discussed below, the composite (time-averaged) velocity and density distribution maps afford new insight into flow mixing and density

stratification in the gravity current; and further suggest a new mechanism to explain the subdued topography of seafloor bedforms through self-limited development.

**Flow structure**. Figure 3 shows the time average downstream velocity and distribution of relative excess density in the Black Sea gravity current. The flow is highly turbulent, with a Reynolds number $Re \sim 2.5$ million, and subcritical, with a bulk Froude number $Fr \sim 0.6$ (Fig. 3), calculated from directly sampled velocity and density data. The velocity maximum acts like a free-surface within the subcritical flow, dipping over the crest of the low amplitude bedforms[2] (Fig. 3a); correspondingly the Froude number varies in phase with the bedform topography (inset Fig. 3b). In both the upper and lower shear layer the flow velocity decreases rapidly from its maximum value. This is in marked contrast to standard gravity current models for two main reasons firstly, the occurrence of a concave down, not concave up, velocity profile below the velocity maximum and secondly the linearly exponentially decreasing, rather than Gaussian, form of the upper shear layer (as highlighted by the rapid decrease in flow velocity just above the velocity maximum, see insets in Fig. 3b) in this subcritical flow. The integral flow depth[7] is approximately constant, $H \sim 8$ m.

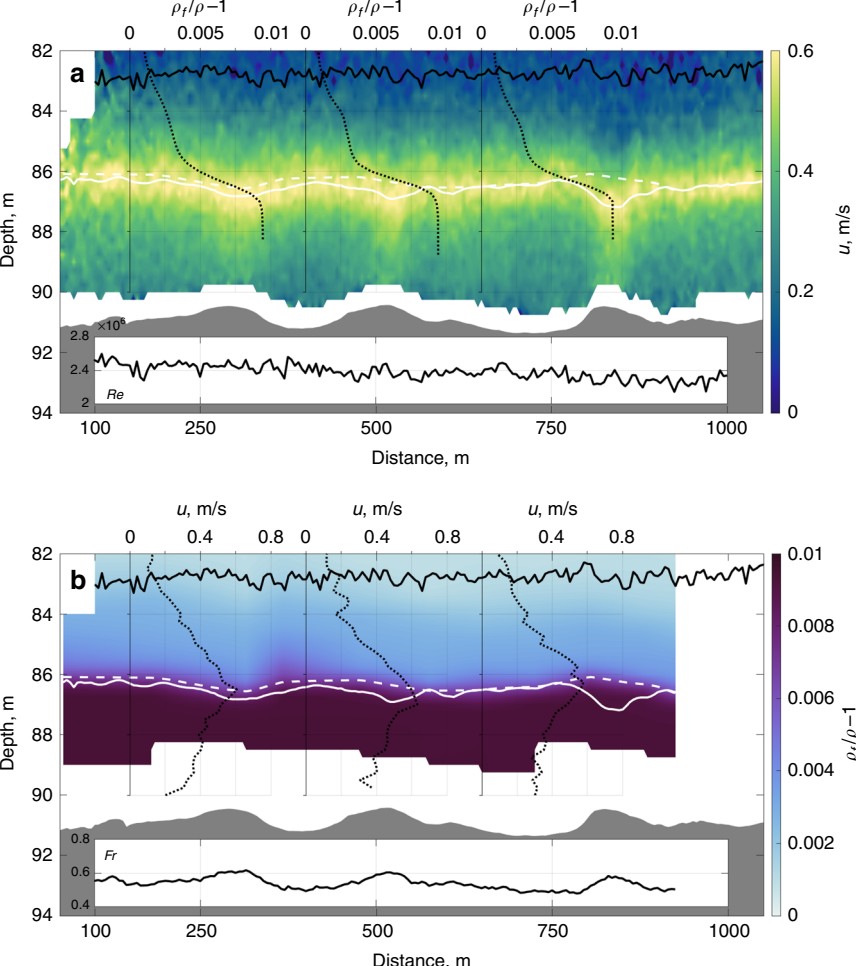

**Fig. 3** Downstream time-averaged flow velocity and excess density maps. Downstream flow velocity (**a**) and excess density (**b**) above the seafloor bedforms (shaded grey). Insets in (**a**) and (**b**), respectively, depict the depth averaged Reynolds and Froude numbers (see methods). Dotted black lines respectively highlight spatial variability in the relative buoyancy, $\rho_f/\rho - 1$ (**a**), and the downstream time-averaged flow velocity, $u$ (**b**), where the velocity maximum is co-located with a sharp density gradient. Solid white lines depict the velocity maximum, along a contour of zero gradient, dashed white lines depict the maximum density gradient and solid black lines the integral flow depth[7]

The flow exhibits a strong density gradient, where relative density decreases by a factor of four within one tenth of the flow height, strongly correlated with the height of the flow velocity maximum (Fig. 3b). In a saline gravity current this organised structure is surprising; since the existing flow paradigm suggests that such a gradient would be smoothed by turbulent diffusion given the flow has travelled ~35 km from source. The sharp density gradient in the flow suggests that there is a significant barrier to mixing between the lower and upper shear layers. The reduction of shear-driven turbulence production at the velocity maximum, i.e. a slow-diffusion zone, has in the past been postulated as a mechanism to constrain mixing between the lower and upper shear layers[8,45]. However, the assumption of a reduction in turbulence through decreased shear production neglects the role of advection and diffusion of turbulence that enables turbulent mixing across an internal flow velocity maximum[24,46,47]. Thus, a slow-diffusion zone mixing barrier likely only arises in strongly depositional flows, where the interplay between reduction of turbulence production at the velocity maximum and sedimentation of particulate material drives run-away stratification-induced turbulence dampening[15]. Moreover, it is improbable that a slow-diffusion zone could develop, or persist, in real-world flows, where the flow is subject to three-dimensional mixing and fluctuations arising from topographic forcing and intermittent flow[4,10,13,47]. Alternatively, sharp internal gradients are generated by dilute flow shed from the head of a gravity current[1]. However, this is not applicable in the quasi-permanent Black Sea gravity current[22]. Therefore, the presence of a sharp internal gradient in flow density (see insets Fig. 3a), correlated to the velocity maximum, is in

contrast with standard models of slowly-varying density distribution in saline, or low settling-velocity sediment-laden, gravity currents (Fig. 1)[11,18].

**Self-organised gravity currents**. To elucidate the nature of the divergence between our field data and standard gravity current models[48,49], respectively, a comparison of a shallow, quasi-continuous stratified gravity current that has travelled far from source to models based on flows of comparatively short duration or development lengths, a standard transformed coordinate system is employed[50–52]. In the transformed coordinate system flow depth, $z$, is centred on the velocity maximum, $z_m$, and is normalized by the integral flow length scale (see methods). This coordinate system enables spatial averages to be made of the velocity and excess density profiles unweighted by the position of the velocity maximum or scale of the flow (Fig. 4). Figure 4 reveals that the distribution of average velocity and density profiles in the Black Sea flow (Fig. 4c) in fact have more in common with those of oceanic and atmospheric zonal jets (Fig. 4d, e) than previously documented gravity currents (Fig. 4a, b). The Black Sea gravity current is characterised by self-organisation of the flow. The self-organisation of the Black Sea gravity current can be explained completely by analogy to the dynamics of zonal jets. Whilst zonal jets are formed and forced by Rossby waves, gravity currents are driven by a density difference from their surroundings. Thus, an obvious candidate for self-organisation of gravity currents are dispersive internal gravity waves[53], as in the QBO and Phillips effect.

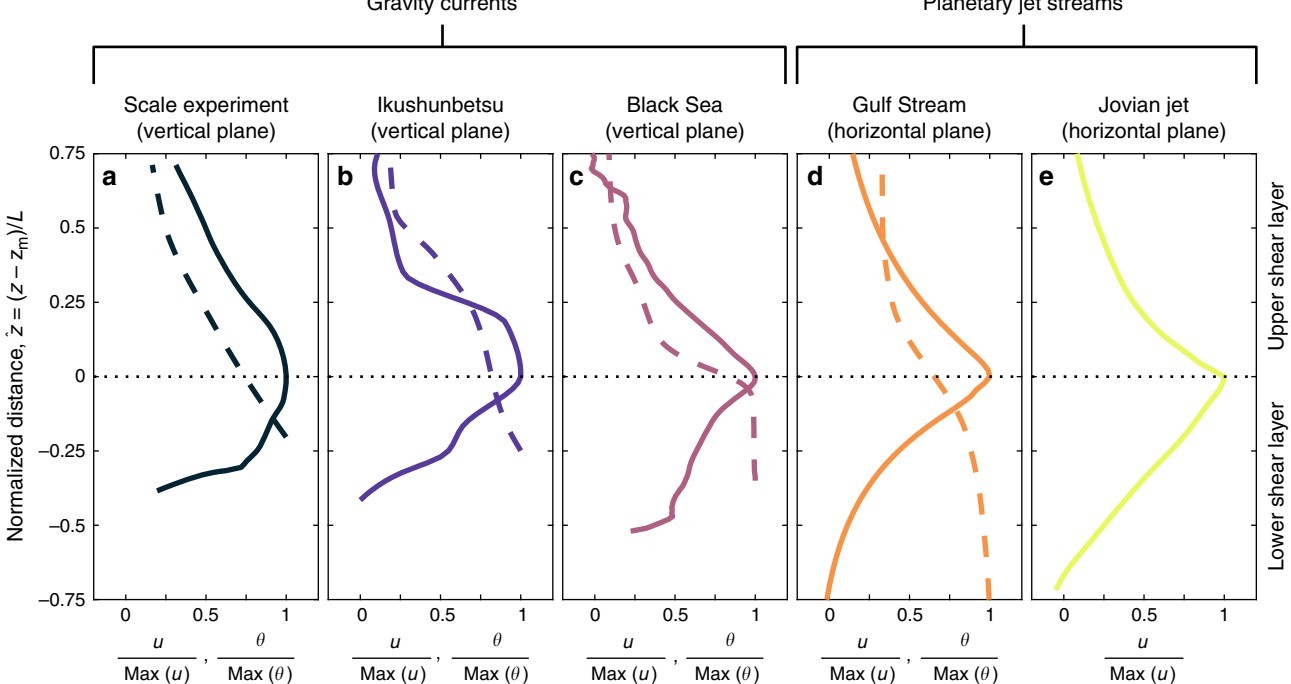

**Fig. 4** Comparison of time-averaged velocity and density profiles across a range of environmental flows with and without sharpened-jet-like structures. Normalized velocity and scalar transport profiles of: **a** a saline gravity current experiment, Exp. 23[11]; **b** a sediment-laden gravity current in the submerged Ikushunbetsu river valley[48]; **c** the mean Black Sea gravity current investigated herein; **d** the North-Atlantic Gulf Stream at 70°W[49,50]; **e** a high speed Jovian atmospheric jet stream at 24°N[51]. In (**a–e**) streamwise velocity, $u$, is denoted by solid lines, and the concentration of the scalar quantities transported by the flows, $\theta$, are shown by dashed lines; the plane of measurements is specified with respect to gravity; and dashed black lines denote the lower–upper shear layer interface; and cross-stream distance, centred at the velocity maximum, $z_m$, is normalized by the flow integral length scale, $L$ (see methods). In (**a–c**) the scalar transport term is $\theta = \rho_f/\rho - 1$ and in (**d**) $\theta$ is the sea surface temperature. The velocity and scalar transport profile of the Black Sea gravity current (**c**) is conspicuously different from existing experimental and field-data based models of density-driven flows (**a, b**), but strongly resembles the self-sharpened profiles of oceanic and atmospheric jets (**d, e**)

Figures 3 and 4 show that the stratified Black Sea gravity current has a sharpened-jet-like velocity profile with coincident density layering (Fig. 4c). We hypothesize that this self-organisation arises as a result of the flow over the low-relief bedforms. The flow over the bedforms drives disturbance of the density stratified fluid from a neutral buoyant level, thus generating internal gravity waves[2,53], whilst vortex shedding from bedform crests enhances coherent eddies in the flow[54]. These internal gravity waves result in momentum transport to a critical layer near the velocity maximum, where wave breaking and momentum absorption locally accelerates the flow in a fashion analogous to the well-postulated models of the sharpening of zonal jets cf.[26–35]. Internal gravity wave driven anti-diffusive momentum transport contrasts to the standard diffusive mixing gravity current model (Fig. 1), but offers a robust explanation for the self-sharpened, concave up nature of the velocity profile that is clearly evident in the lower shear layer. Extant empirical evidence details interfacial instabilities that result in internal coherent eddies and associated internal gravity waves[12,22], these explain the sharpened velocity profile observed in the upper shear layer of the flow (see Fig. 3 and also compare the Gaussian-slow and linear-rapid exponential decay of the velocity profile from the velocity maximum in Fig. 4a, c, respectively). Measurements of turbidity current velocity in Monterey and Hueneme canyons also show near-linear decay profiles of the upper shear layer[52], previously interpreted as a product of supercritical flow[18] (see earlier discussion). Alternatively, these profiles are plausibly explained by self-organisation via internal gravity wave forcing[26]. Density layering is also explained in a similar fashion. Gravity wave breaking, enhanced by coherent eddies[55], homogenises PV. This results in a PV inhomogeneity at the velocity maximum, that forms an eddy transport barrier (rather than a slow-diffusion zone) capable of maintaining sharp internal density gradients over long distances. Strong PV gradients provide a restoring mechanism, i.e. gravity wave elasticity, which is recognized in the flexible boundary-like behaviour of the velocity maximum (Fig. 3).

In zonal jets principal flow shear is normal-to-gravity (Fig. 1b), expressed in the vertical component of the vorticity vector[24]. As shear is coincident with vertical density stratification this enables the use of simplified two-dimensional flow models[26–35] to describe PV conservation. In contrast in gravity currents, whilst density stratification is again the vertical plane, principal flow shear is parallel to gravity (Fig. 1a), and is expressed in the cross-stream component of the vorticity vector[24]. Thus, requiring the use of a generalised three-dimensional flow model, i.e. Ertel's PV theorem[28]. However, assuming a two-dimensional gravity current, with no cross-stream density variation or flow, the PV (the dot product of density stratification and absolute vorticity) has the trivial solution of zero everywhere. Thus, in a two-dimensional gravity current there can be no PV inhomogeneity and no eddy transport barrier. The field-data (Fig. 3), however, imply an eddy transport barrier. In turn, this implies that three-dimensional velocity and density gradients within gravity currents have a key effect on flow dynamics, contrasting quasi-two-dimensional flow models commonly used[1,17–19]. The exception to this being where weak lateral baroclinic sources enable self-organisation of a quasi-two-dimensional flow; yet even here flow is three dimensional, with a lateral component along the density gradient. Indeed, velocity and density profiles of gravity currents are rarely laterally homogeneous[22,56]. It is expected that these mechanics translate to other gravity currents where topography, mixing and shear enable the generation, propagation and breaking of internal gravity waves. These processes likely also have applicability to wave-turbulence interactions, and mixing, in environmental flows.

**Bedforms.** All Earth surface flows have the capability to sculpt boundaries composed of mobile particulate material. Commonly, flow interaction with mobile boundaries produces wave-like sinuous deformation, i.e. bedforms[54]. Bedforms are a critical component of many natural sedimentary environments, controlling flow and sediment discharge[57]. In many sedimentary environments bedforms impart significant roughness to flows, via their pronounced relief; however the seafloor is often relatively smooth compared with the bed topography typically observed within riverine or estuarine environments. Despite normally being much larger than their terrestrial analogues, seafloor gravity currents typically form long wavelength, low amplitude high aspect-ratio bedforms[58]. The reason for the formation of low-relief seafloor bedforms has hitherto remained unclear, but the data presented in Fig. 5a–d, in which the time-averaged composite velocity profile is separated into stoss, crest, lee and components, defined by ± 1 standard deviation from the mean sloped bed (Fig. 5e), afford some insight into this problem. Whilst the four bedform components have similar velocity profiles, the mean shear in the lowermost 2 m of measured flow varies strongly across the bedforms (note that shear within ~1 m of the bed cannot be computed directly due to the absence of reliable flow velocity data in that region). The mean shear is lowest over the trough and stoss sides of the bedforms, before increasing by 50% over the crest and 25% over the lee sides of the bedforms (Fig. 5). This is attributed to the dynamical behaviour of the velocity maximum, due to both stable density layering and gravity wave elasticity. Acceleration over the bedform crest and deceleration over the bedform troughs from the flexible velocity maximum is thus a result of subcritical lower shear layer flow compression and expansion[2]. Flow acceleration and deceleration results in an enhanced ability to erode material from the crest and a diminished ability to erode material from bedform troughs. That bedform evolution is self-limited by the dynamics of the near-bed velocity maximum, not the depth of the entire flow, may therefore explain the enigmatic high aspect-ratio of bedforms sculpted by seafloor gravity currents.

## Discussion

The velocity and density data from the Black Sea flow demand a fundamental reappraisal of traditional models of the dynamics of stratified seafloor gravity currents, where internal gravity waves may develop (Fig. 6). Our data strongly suggest that roughness imparted by low-relief bedforms results in the formation and maintenance of a jet-like profile with a self-sharpened locally increased flow velocity (Fig. 6b, c), in contrast to the current paradigm (Fig. 6a). These new observations are explained by a robust theoretical framework, recently advanced in fundamental fluid dynamics[26,27], which implies that increased velocity arises from dispersive internal gravity wave transport of momentum to a critical layer within the flow where it is absorbed. Internal gravity waves imply statistically unstable flow, where mean conditions vary spatially and temporally; this contrasts with current long-duration gravity current models that for simplicity assume simplified or statistically stable, i.e. constant, mean flow[16,59]. Moreover, the theoretical framework implies internal gravity wave mixing of PV at the critical layer results in a stabilised eddy transport barrier, preventing transport across the velocity maxima[32], as evidenced by the field-scale measurements herein, leading to a two-layer flow. This two-layer flow acts to maintain the concentration, and thus momentum in the lower-part of the flow, through restricting transport of material upwards across the interface, and simultaneously limiting entrainment of ambient fluid from the upper flow.

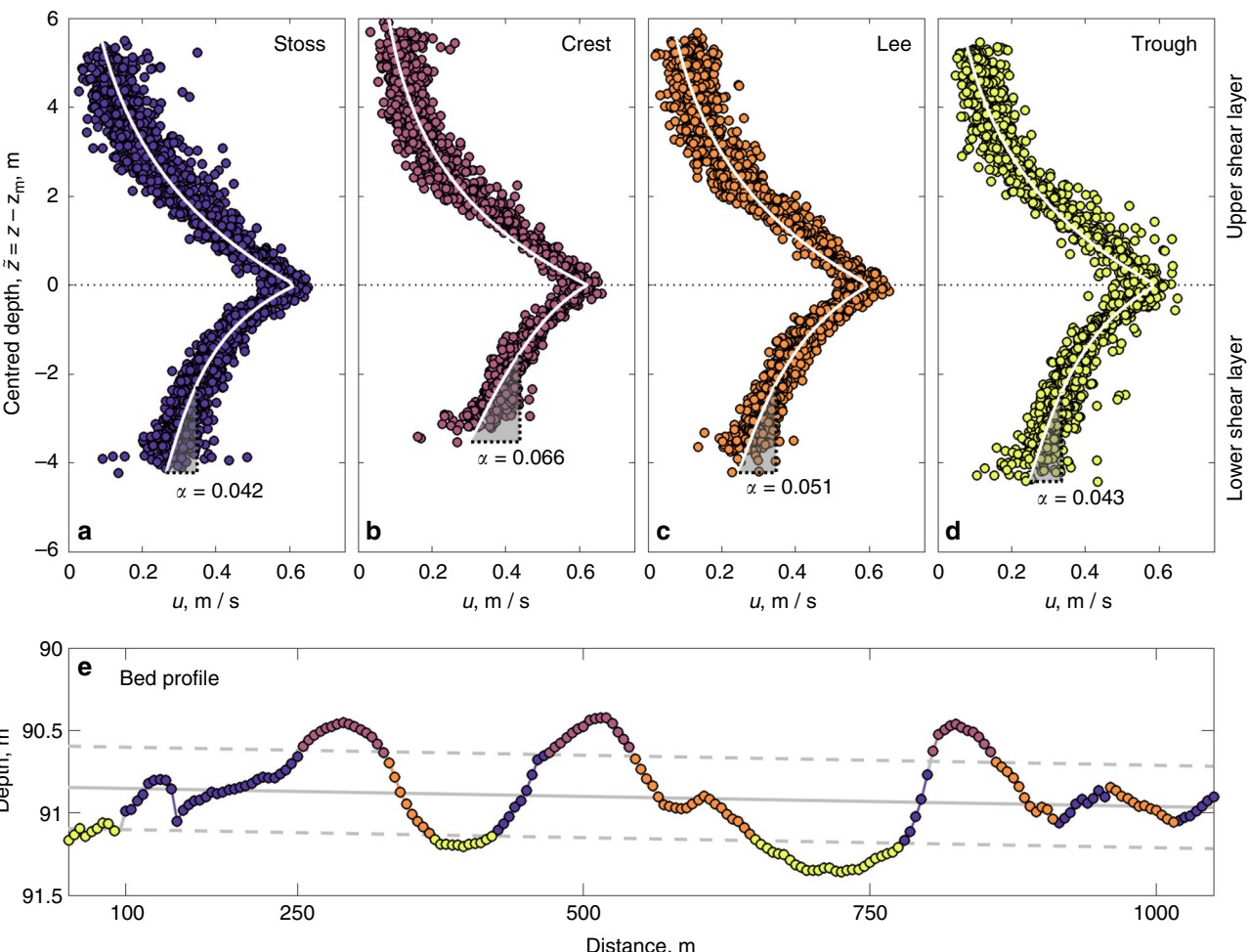

**Fig. 5** Velocity profiles. Vertical profiles of streamwise flow velocity, $u$, as a function of centred flow depth of the stoss-side (**a**), crest (**b**), lee-side (c) and trough components (**d**) of the three channel bedforms (**e**). In (**a**–**d**) the white lines denote least squares two-term linear-exponential curves of best fit (see methods), made above and below the velocity maximum; $\alpha$ denotes the vertical derivative of velocity, s$^{-1}$, over the lowermost 2 m of the flow. In (**a**–**e**) symbol shading denotes the four components of each bedform, (**a**–**d**). In (**e**), the mean bed depth is depicted by the solid grey line; with dashed grey lines denoting a one standard deviation confidence interval. Within one standard deviation of the mean bed depth stoss-side and lee-side are defined by local slope; the trough and crest are defined by the regions outside of one standard deviation of the mean bed depth

The mixing process described (see Fig. 6) results in very strong flow stratification, in turn aiding long-term maintenance of the flow, and thus the run-out distance of a flow cf.[16]. As stratification, and thus the potential for dispersive gravity waves, is always generated through ambient fluid entrainment or gravitational settling the observationally validated theoretical framework is extendable to all gravity currents where dispersive waves operate over sufficient time- and length-scales to modify the mean flow[26,35]. Our results bring in to question whether extant experimental and numerical studies of gravity currents[7–18] are of sufficient spatial and temporal resolution and scale to capture the evolution of flows under internal wave forcing, with previous research showing significant discrepancy between theoretical and real-world flow dynamics[16]. Flow evolution over large scales is important, since the long run-out distance of gravity currents in submarine channels has proven to be an enigma[16,18,59–61]. The jet-sharpening model thus paradoxically predicts that increased bed roughness likely enhances net flow transport. Furthermore, the changes in the flow result in a negative feedback to the bedforms that may lead to optimisation of bed roughness with respect to total flow transport.

The positive influence of bed roughness has previously been observed in terms of drag reduction. Drag reduction is generated by small-scale (height ~1–10 μm) roughness (topography) induced modification of boundary layer flow, for instance flows across shark skin[62] and golf balls[63]. Here we show that the self-sharpening of seafloor gravity currents results in enhanced flow velocities at topographic (bed roughness) length scales of approximately one metre, 5–6 orders of magnitude greater than in boundary-layer drag reduction. Increased velocity results in increased applied force on objects immersed in the flow, increasing the geohazard-risk seafloor gravity currents pose to marine infrastructure[64,65].

These new observations require evaluation of the time and length scales over which internal gravity wave forced flows converge to a pseudo-steady state. Further, the conditions supporting internal gravity wave development needs parameterization and three-dimensional and statistically unsteady flow processes require analysis. More generically the data presented are evidence for the need for further experimental and numerical quantification of turbulent mixing processes in sediment-laden, stratified Earth surface flows and its quantification in terms of conservation of circulation (i.e. Ertel's vorticity theorem). Whilst further work is required to address these outstanding questions, our new data and supporting theoretical analysis presented here open a new field on the role of dispersive waves (arising from boundary layer

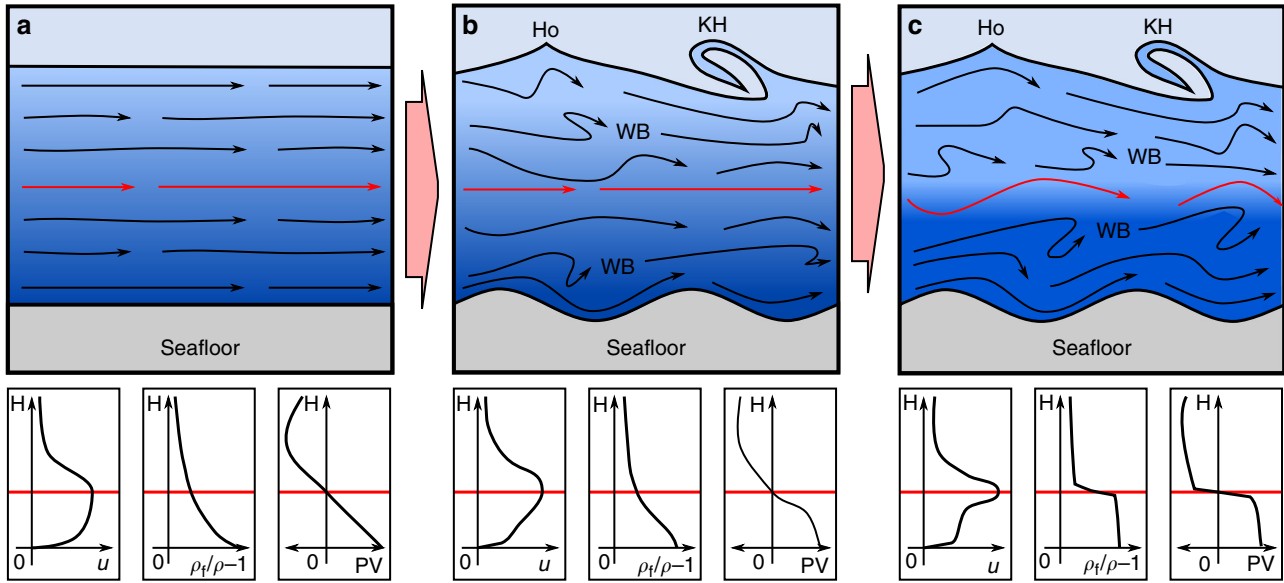

**Fig. 6** Sketched evolution of jet-sharpening in a gravity current, bounded by a lower seafloor interface and an upper flow—ambient fluid interface. Smooth flow boundaries result in predominantly parallel streamlines (**a**), with some internal variation due to turbulent fluid motion and coherent eddies shed into the flow, see Fig. 1. Deformation of the lower and upper flow boundaries (**b**), e.g. by bedforms or the onset of Holmboe (Ho) or Kelvin-Helmholtz (KH) instabilities, enhances vertical movement within the flow. Vertical fluid motion results in the formation of internal gravity waves due to the buoyancy restoring force arising through density stratification. Sheared by downstream flow, gravity waves generate up-velocity-gradient momentum transport. At a critical layer, on the gravity current flanks, shear results in gravity wave breaking (WB). Wave breaking results in the irreversible transfer of momentum into the flow, and mixing of flow density and potential vorticity (PV). Progressive momentum transfer accelerates the gravity current core (**c**), a process ultimately limited by viscous and turbulent dissipation. Wave breaking also results in the homogenous mixing of density and PV either side of the velocity maximum. The sharp PV gradient stabilises the velocity maximum, acting as a restoring mechanism i.e. gravity wave elasticity, and as a turbulent eddy barrier to mixing between layers. Thus, for flow over bedforms, the quasi-rigid layer containing the velocity maximum responds as a free-surface (**c**). In (**a**–**c**) subplots denote evolution of velocity, $u$, relative density, $\rho_f/\rho - 1$, and PV, based on vertical gradient of downstream velocity. Decreasing flow density is denoted by shading, from dark to light blue; streamlines are denoted by black arrows; red lines denote location of velocity maximum

roughness, topographic and interfacial waves or instabilities) on density stratified Earth surface flows.

## Methods

**Field-data collection.** Flow velocity was collected using a downward looking Teledyne RDI 1200 kHz Acoustic Doppler Current Profiler (ADCP), capturing three-component velocity data with a vertical bin size of 0.25 m, deployed from the AUV Autosub III[21]. Sixteen transects of three-component velocity data were collected within 6 h, between 11.20 h and 17.17 h on the 5th July 2013, over a 1 km stretch of seafloor spanning three bedform features (Fig. 2). Transects were collected in sets, with the AUV deployed progressively closer to the seafloor at heights ~10 m (6 repeats), ~7 m (6 repeats) and ~5 m (4 repeats) above the mean bed depth. This was done to produce enhanced resolution of the near-bed flow. Bed topography was collected from the R/V Pelagia using a RESON Seabat 7125 Multibeam echo sounder (MBES) with coupled motion and position provided by a Trimble Applanix POSMV 320. Bathymetric soundings were processed within the CARIS-HIPS 10.1 engine, where data were corrected for sound velocity variation, and tides. The soundings were gridded to 1 m resolution within ArcGIS to create a raster based digital elevation model. A straight-line master transect of best fit was derived from linear least square regression of the sixteen individual AUV transect paths. Raw velocities were adjusted for AUV motion, corrected for true position and filtered following the methodology laid out by Dorrell et al.[22]. Specifically, data below the maximum ADCP backscatter intensity or within the blanking distance $y_b$ of the seafloor were discarded, where $y_b = y_a \sin^2\vartheta$[66] given the altitude of the AUV with respect to the seabed, $y_a$, and the angle of the profiling beam $\vartheta = 20°$. For each individual AUV transect, velocities were mapped using orthogonal projection on to the master transect and interpolated onto a 5 m by 0.25 m (downstream by vertical) mesh using Matlab's[TM] 2013a linear griddata function. Within 15 h of the AUV deployment, 11 vertical CTD casts were taken on the 6th July 2013 using a Seabird 19 profiler deployed from a stationary research vessel that had dynamic positioning (R/V Pelagia). From CTD measurements, water density, $\rho$, was derived using the UNESCO formula[67]. As per the velocity profiles, flow density and density gradients were mapped on to the master transect using an orthogonal projection and interpolated onto the same 5 m by 0.25 m rectilinear mesh.

**Flow parameterization.** The composite velocity map was based on the average velocity from all sixteen transects. The local bed depth, $\eta$, was similarly specified by

the mean of the 16 ADCP bottom tracks. To calculate bulk flow parameters at each point along the master transect, the composite dataset was interpolated to a no-slip boundary condition on the seafloor using Matlab's[TM] 2013a cubic interpolation function[68] and density was assumed to be constant between the lowest recorded measurement and the bed[22]. A reference height, $h$, was defined, where $\eta + h = -82$, the upper limit of flow sampled by the AUV. From the composite flow velocity, the ratio of inertial to viscous forces, the Reynolds number, was calculated along channel using

$$\mathrm{Re} = \frac{\int_\eta^{\eta+h}\sqrt{u^2 + v^2}\,\mathrm{d}z}{\nu}, \tag{1}$$

where the Black Sea flow, with mean temperature of 7.5 °C and density 1.020 kg m$^{-3}$, has an average kinematic viscosity[69] $\nu = 1.33 \times 10^{-6}$ m s$^{-2}$. In a similar manner, the ratio of inertial to gravitational forces, the Froude number, was calculated with

$$\mathrm{Fr} = \frac{\int_\eta^{\eta+h}\sqrt{u^2 + v^2}\,\mathrm{d}z/h}{\sqrt{g\int_\eta^{\eta+h}\left(\frac{\rho_f}{\rho} - 1\right)\mathrm{d}z}}, \tag{2}$$

where $g$ denotes gravity whilst $\rho$ is the absolute and $\rho_f$ the local flow density. We note that Froude number is a bulk property and thus is not entirely appropriate to highly stratified flows[22,66]. After Ellison and Turner[7], the integral flow-depth is

$$H = \left(\int_\eta^{\eta+h}\sqrt{u^2 + v^2}\,\mathrm{d}z\right)^2 / \int_\eta^{\eta+h}\left(u^2 + v^2\right)\mathrm{d}z. \tag{3}$$

The integral length scale of the flow, $L$, is specified solely in terms of the along transect flow velocity, $u$,

$$L = \left(\int_{-\infty}^{\infty}u\,\mathrm{d}z\right)^2 / \int_{-\infty}^{\infty}u^2\,\mathrm{d}z. \tag{4}$$

In the Black Sea gravity current $H \approx L$ as cross-stream velocity is negligible in comparison to downstream velocity. Best fit curves (Figs. 4 and 5) to the velocity profile data, $u_{\mathrm{fit}}$, were made using a two-term linear-exponential model[50], as a

function of normalized depth $z'$,

$$u_{\text{fit}} = ae^{-bz'} + ce^{dz'}, \qquad (5)$$

optimised using Matlab's$^{\text{TM}}$ 2013a exponential fit algorithm using a non-linear least squares method[70].

## Data availability

The authors declare that the ADCP derived velocity data and CTD derived density source data that support Figs. 3a, b, 4c and 5a–e are provided as a Source Data File and are available online at https://www.bodc.ac.uk/data/published_data_library/catalogue/10.5285/7a8bd6b3-f066-31ea-e053-6c86abc00899/.

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

## Acknowledgements

This project was funded by Natural Environment Research Council (NERC) grants NE/F020511/1, NE/F020120/1, and NE/F020279/1. We thank the crew of the RV Pelagia for their assistance with cruise planning and operation, and the Marine Autonomous and Robotic Systems (MARS) engineers from NERC National Marine Facilities for AUV operational support. We thank Steven Tobias, for helpful discussions on an early version of the manuscript, as well as David Dritschel and two anonymous reviewers and Senior Editor Eithne Tynan for their constructive comments which have substantially improved this paper.

## Author contributions

R.M.D. undertook the analysis and developed the theory. R.M.D., S.E.D., D.R.P., J.J., J.P., E.J.S., R.W., E.O. and D.T. collected and processed the field data. All authors contributed to drafting and editing the manuscript and co-designed the field-study. J.P., D.R.P., S.E.D. and R.W. jointly conceived the supporting grant(s).
