## [peer review file · Nature Communications]

Reviewer #1 (Remarks to the Author):

The authors present a model for self-sharpening gravity currents that is in sharp contrast to generally accepted models based on turbulent diffusion of scalar properties.

It is in general well written, the figures well prepared, and it is mostly well-referenced. However, the main fluid mechanical arguments are introduced far too early, and as a result are repetitive since they are reprised at some length later in the paper. It isn't clear who the audience is meant to be, and much of it was not in my field, so I can't comment in detail. Nonetheless I saw what seem to be some inconsistencies, such as the suggested lack of large scale coherent turbulent structures over a flat bed even when the Reynolds number is over 106.

I am left with the feeling that the authors believe that this has far wider application than is justified by modern observations on turbidity currents (e.g. Xu 2010). Surely turbulent diffusion of sediment away from the bed is a necessary process for the maintenance of the suspension.

I recommend publication, but with some caveats, namely that the authors address these reservations, and also those in the attached document that refer to specific places in the text by line number

Reviewer #2 (Remarks to the Author):

This paper demonstrates that the classical paradigm of oceanic gravity currents may not always apply, and indeed may be incorrect. The authors demonstrate, using unprecedented high quality measurements and analyses of the near sea-bed density and velocity fields, that gravity currents may rather behave like sharpened zonal jets in planetary atmospheres (and also like the Gulf Stream and other major ocean currents, though this is not noted in the paper). I believe this is a remarkable result of great insight, and I strongly support publication.

There is one issue that remains somewhat unclear; in lines 243-252 the authors correctly point out that the PV of a two-dimensional (x-z) flow is zero, and thus three-dimensional velocity and density variations are needed to explain the existence and sharpening of the PV in the gravity current. But I do not think this rules out this process occurring in a two-dimensional model of flow over a weakly varying bedform: density gradients may steepen by gravity-wave breaking and mixing, and the location of sharpest density gradients may naturally (on account of its greater stability) "attract" a more coherent flow less affected by turbulent mixing. In other words, flow along the sharp density gradient may not be as easily broken down as flow away from this gradient, and the eddy fluxes may

in fact support momentum deposition in the region of sharp density gradients. This would imply that the horizontal vorticity perpendicular to the plane of the assumed two-dimensional flow acts somewhat like PV in jet sharpening, even though this vorticity is subject to baroclinic sources (horizontal density variations). I believe this is a possibility and one that cannot yet be ruled out.

David Dritschel

University of St Andrews

6 September 2018

Reviewer #3 (Remarks to the Author):

A review of "SELF-SHARPENED GRAVITY CURRENTS SELF-SHARPENING INDUCES JET-LIKE STRUCTURE IN SEAFLOOR GRAVITY CURRENTS"

The paper is well written and very interesting. It described fascinating data on gravity currents from the Black Sea. The data is well described and is of significant importance to be published in Nature Communications. However, I do not agree with the theoretical explanation which has no mathematics to back it up. It argues by analogy primarily with planetary scale phenomena. I believe that most of the speculation should be removed and that the paper would then be suitable for publication. Better yet would be a theoretical calculation, like in the McIntyre, that actually shows that their proposed mechanism is correct, though since I think it is wrong I doubt this is possible.

The use of the word "dissipation" instead of "diffusion" does not add to the confidence of the reader in the theoretical understanding of the authors.

I've read reference (25) McIntyre, M. E. (2008). "Potential-vorticity

inversion and the wave-turbulence jigsaw: some recent clarifications. *Advances in Geosciences*, 15, 47-56."

And I have trouble seeing how it is an explanation for possible sharpening in gravity currents. It is concerned with large scale rotating flows where the key idea is the invertibility of potential vorticity, that is at any time the potential vorticity hold all the information about the flow and the velocity fields and density fields can be calculated from it. Rotation is only marginally important for only the largest gravity currents and the "invertibility" of potential vorticity does not hold.

for example the McIntyre says "A full understanding of such phenomena can scarcely be arrived at without the conscious or unconscious use of the PV "invertibility principle" or something equivalent (e.g., Hoskins et al., 1985 & refs). It is an essential component of the conceptual "Lego set" one needs."

I do not see therefore how this paper can explain the phenomena discussed in this paper.

line 18 "isotropic turbulent fluctuations" The turbulence is non-isotropic in regions influenced by the boundaary

Figure 1 The figure says "dissipation of momentum" but momentum is conserved; It cannot be "dissipated". In this and many other places (i.e line 66) in the manuscript, dissipated should be changed to diffusion.

Line 71 "the self-organisation of turbulence, by dispersive internal waves, results in up-velocity-gradient momentum transport"

This needs a reference. I find this expanation quite hard to believe

Line 72 "whilst irreversible wave breaking transfers wave energy into the mean flow." This also doesn't seem right to me. wave breaking will increase the turbulence intensity but won't increase the mean flow.

line 80 "negative viscosity" I would say that this is highly controversial and any model that includes things in this form is highly phenomenological and technically speaking is ill-posed and admits no grid independent solutions.

We would like to thank the reviewers for their detailed commentaries on the manuscript. We have addressed (as highlighted in blue below) all the comments of each of the reviewers and in our response below we detail the action we have taken in relation to each point. The line numbers refer to location of the changes we have made in the new (revised) manuscript, where we have tracked all the changes to allow full transparency in the next round of review.

Response to Reviewer #1:

The authors present a model for self-sharpening gravity currents that is in sharp contrast to generally accepted models based on turbulent diffusion of scalar properties. It is in general well written, the figures well prepared, and it is mostly well-referenced. However, the main fluid mechanical arguments are introduced far too early, and as a result are repetitive since they are reprised at some length later in the paper.

We understand fully this point. In the original paper we were caught between needing the introduction to cover some basic fluid mechanics of the different flows contrasted in the paper, particularly as the paper bridges traditional research disciplines (mathematics, atmospheric sciences and earth surface processes), and the need to provide a more detailed synthesis section that provides the theoretical underpinning. Detailed further in the responses below, we have made some amendments that we hope strike a more appropriate balance in this regard, including:

1. Rewording the abstract to highlight the multidisciplinary nature of the research presented in the manuscript.
2. Where possible simplifying, and shortening, introductory text covering both gravity current and zonal jet dynamics.
3. Revising the literature reviewed to make sure it fully support arguments presented.
4. Further highlighting the assumed fundamental differences between physical processes of different environmental flows.
5. Clarifying how physical processes may be translated between different environmental settings.
6. Breaking the manuscript down into distinct introduction, results and discussion sections.

It isn't clear who the audience is meant to be, and much of it was not in my field, so I can't comment in detail.

We have revised the abstract to highlight that the paper is of particular interest to researchers interested in gravity current dynamics, wave-turbulence interaction and material transport process in a wide range of environmental flows, which cover a broad range of fields and applications *"These processes likely have broader application, for example to wave-turbulence interaction, and mixing processes in environmental flows"*.

We have also made this point clearer in the main text. We hope that this will result in the paper receiving more attention and thank the reviewer for this steer.

Nonetheless I saw what seem to be some inconsistencies, such as the suggested lack of large scale coherent turbulent structures over a flat bed even when the Reynolds number is over 10^6 .

We have revised the text to make it clear that we do not suggest there is a lack of large-scale coherent turbulent structures over a flat bed at high Reynolds numbers. We have emphasised the text to state that the presence of bedforms actually (L231): *"enhances coherent eddies in the flow"*

I am left with the feeling that the authors believe that this has far wider application than is justified by modern observations on turbidity currents (e.g. Xu 2010). Surely turbulent diffusion of sediment away from the bed is a necessary process for the maintenance of the suspension.

Turbulent diffusion is a model to describe the role of "random" motion in non-fully resolved models of fluid flows and thus the processes for maintenance of sediment suspension. We have highlighted the use of such models at L55-58 *"Thus, where models do not fully resolve turbulent fluid motion, the frictional turbulent diffusion of momentum, and analogously diffusion of concentrations of quantities transported by the flow, are*

parameterized by a positive [Kraichnan, 1987] eddy diffusivity model (the second of two common key assumptions on gravity current flow dynamics)."

In further response to the reviewer's comment we note that it has previously been established that turbulent diffusion is fundamentally flawed when turbulent motion results in mixing at scales larger than flow variable length scales (e.g. Nielsen & Teakle, Physics of Fluids, 2342, 2004). Further, in the manuscript we go on to show that: i) internal gravity waves result in large scale mixing processes; and ii) large scale mixing processes result in self-organisation of the flow with analogy to existing theory describing atmospheric flows.

Xu (2010), whilst of a much lower resolution than the present work, in particular with far more limited data in the lower shear layer, does show similarities with the present work in the outer shear layer (see response to point I. 211 below). Thus evidence from modern turbidity currents (Xu, 2010) that is otherwise enigmatic can be explained by the present work. In addition to the present work having applicability to turbidity currents, we have modified the text to make it clear that there are also broader implications for wave-turbulence and interaction, as well as mixing processes in other environmental flows.

We have now clarified these points to address the reviewer's comments by amending the abstract L22-23: *"These processes likely have broader application, for example to wave-turbulence interaction, and mixing processes in environmental flows"*, as well as noting this in the main text.

We further discuss the Xu (2010) data in the text L240-242: *"Field measurements of turbidity currents in Monterey and Hueneme canyons also show near-linear decay profiles of the upper shear layer [Xu, 2010], although hitherto there has been little explanation of why this differs from extant Gaussian quadratic decay models [Kneller et al., 1999]"*.

I recommend publication, but with some caveats, namely that the authors address these reservations, and also those in the attached document that refer to specific places in the text by line number

We thank the reviewer for his/her constructive comments and have significantly revised the manuscript to address them as detailed both above and below.

Fig 1a does not represent either the classic experimental and numerical view nor the measured currents in, e.g. Monterey Canyon (Xu and others) which they closely resemble.

The authors thank the reviewer for his/her constructive feedback. The authors have amended the Figure to note that it is a schematic diagram and that the relative size of the lower and upper shear layers of the flow are not drawn to scale. (L84-85): *"The size of the lower and upper shear layer are not drawn to scale, varying with flow and environment conditions"*.

We have also adjusted the main text (L47-48): *"Although the relative size of the upper and lower shear layers vary with environment and flow conditions [Sequerios et al., 2010], the velocity maximum is often located close to the bed [Azpiroz-Zabala et al., 2017], constraining resolution of flow dynamics in the lower shear layer [Azpiroz-Zabala et al., 2017; Kneller et al, 1999; Kneller & Buckee, 2000; Sequerios et al., 2010; Negretti et al., 2015; Ho et al., 2017]"*.

I. 49 I think Kneller et al., 1999 should be references somewhere around line 49 (or even before)

Amended as requested.

I.55 What they have just described in the text is nothing like what is shown in Fig 1a, which is close to what Sequeiros describe in what they refer to as subcritical flows (which has rarely if ever been described before)

As per Figure 1 the authors have amended this, and the accompanying main text, to note that the relative size of the lower shear layer is dependent on environment and flow conditions. We now provide a reference to Kneller et al., 1999 to address this concern directly L47-48: *"Although the relative size of the upper and lower shear layers vary with environment and flow conditions [Sequerios et al., 2010], the velocity maximum is often located close to the bed [Azpiroz-Zabala et al., 2017], constraining resolution of flow dynamics in the lower shear layer [Azpiroz-Zabala et al., 2017; Kneller et al, 1999; Kneller & Buckee, 2000; Sequerios et al., 2010; Negretti et al., 2015; Ho et al., 2017]"*.

I. 57 This is not really the appropriate reference. Again, maybe Kneller et al., 1999 – or at least something earlier than 2016.

Amended as requested.

The description of data in lines 127 onwards make no reference to fig 3 until line 162. Sea floor gradient?

Figure 3 is referred to in Line 140 in the original manuscript (L142 in the revised manuscript). To address this concern we now reference seafloor gradient in the text at L129-130: “*The seafloor over the bedform region dips downstream at a mean gradient of 1.2×10^{-4} m/m.*”

Fig 3 does not convincingly show the downstream concavity in velocity profiles for either the inner or outer shear layers that are shown in fig 1b. Why not include the composite profiles in Fig. 5 (which make the case far more clearly) into this figure?

The transects are useful as they show the spatial variability within the flow, and that the velocity maximum tracks the sharp internal density gradient – this would not be as evident from composite profiles (Fig. 5). We elect to retain fig 3 because of this, however we have amended the figure caption to ensure that this is clear to the reader at L190-192: “*Dotted grey lines respectively highlight spatial variability in the relative buoyancy, $\rho_f/\rho-1$ (a), and show the downstream time-averaged flow velocity, u (b), where the maximum is co-located with the sharp density interface.*”

I. 157 & 164 Is bulk Froude number meaningful? (See Huang et al, 2009)

This is an interesting point and one on which we have previously published on (Sumner et al., 2013; Dorrell et al. 2016) reinforcing the earlier work of Waltham (2004; Journal of Sedimentary Research, 74, 129-134) and Huang et al. (2009). Despite the limitations we outline and the reviewer highlights, it is a parameter that remains commonly used, and it is typically calculated and used to infer dynamical flow regimes (Kneller et al. 1999, 2016). Indeed we employ it in the manuscript to discuss changing flow dynamics over the seafloor bedforms. We thus prefer to state it herein, noting the limitations through reference to our previous work.

We have modified the text in the methods to state (L403-404): “*We note that Froude number is a bulk property and thus is not entirely appropriate to highly stratified flows [Sumner et al., 2013; Dorrell et al., 2016]*”.

lines 208 to 210 These statements do not accurately reflect the relatively recent work of Sequeiros et al., which is pretty relevant generally

We argue that whilst the work of Sequeiros, and other experimental and numerical studies, have made advances, they are restricted by the low Reynolds numbers and short length scales and time scales of their flows. Whilst the effects of internal wave forcing is evident in the velocity and density profiles in the Black Sea flow, the data are insufficient to determine the length-scales and time-scales over which the flow adapts to the said forcing (cf. Dritschel & Scott, 2011 for evolution of flow forced by Rossby waves). Thus, extant experimental and numerical gravity current research is ill-suited to describe long run-out gravity currents where background forcing slowly modifies the flow. We have revised the summary at the end of the paper, to capture this point (L337-345): “*As stratification, and thus the potential for dispersive gravity waves, is always generated through ambient fluid entrainment or gravitational settling the observationally-validated theoretical framework is extendable to all gravity currents where dispersive waves operate over sufficient time- and length-scales to modify the mean flow [Dritschel and McIntyre, 2008, Dritschel and Scott, 2011]. Our results bring into question whether extant experimental and numerical studies of gravity currents [Ellison and Turner, 1959; Garcia and Parker, 1993; Kneller et al., 1999; Kneller and Buckee, 2000; Sequeiros et al., 2010; Ho et al., 2017; Felix, 2001; Huang et al., 2007; Dorrell et al., 2014; Meiburg and Kneller, 2010; Kneller et al., 2016] are of sufficient spatial and temporal scale to capture evolution of flow under internal wave forcing, with previous research showing significant discrepancy between theoretical and real-world flow dynamics [Dorrell et al., 2014]. Flow evolution over large scales is important, since the long run-out distance of gravity currents in submarine channels has proven to be an enigma [Dorrell et al., 2014, Kneller et al., 2016; Luchi et al., 2018; Bagnold et al., 1962; Parker et al., 1986].*”

I. 211 Isn't this the same approach to normalisation as that used by Xu, 2010?

Yes, we have amended this to acknowledge this is standard (cf. Maxworthy, 1984; Rossby & Zhang, 2001). Interestingly Xu's (2010) data, which resolve the upper shear layer, shows

that a Gaussian model of (quadratic) decay (sensu Kneller et al., 1999) is poor. The fit of turbidity current data yields a nonlinear exponential decay with a mean of 1.3 and a standard deviation of 0.2. This is further evidence for non-standard (momentum) mixing in the upper shear-layer.

The text has been modified to reflect this L240-242: "*Field measurements of turbidity currents in Monterey and Hueneme Canyon also show near-linear decay profiles of the upper shear layer [Xu, 10], although hitherto there has been little explanation of why this differs from the proposed Gaussian quadratic decay model [Kneller et al., 1999]*".

I. 227 Where is the evidence for wave-breaking?

Internal-gravity-wave breaking is inevitable where mean flow equals phase speed of internal waves (cf. Dunkerton et al., 2009). The evidence of wave-breaking and self-organisation is the sharp density interface in, and velocity profiles of, the saline flow some ~35 km from source (note there is no sedimentation acting against turbulent diffusion). The extant theory of jet-sharpening can explain these findings. We refer to the second reviewer (Dritschel) who states that: "*The authors demonstrate, using unprecedented high quality measurements and analyses of the near sea-bed density and velocity fields, that gravity currents may rather behave like sharpened zonal jets in planetary atmospheres [...]*" and "*density gradients may steepen by gravity-wave breaking and mixing, and the location of sharpest density gradients may naturally (on account of its greater stability) 'attract' a more coherent flow less affected by turbulent mixing. In other words, flow along the sharp density gradient may not be as easily broken down as flow away from this gradient, and the eddy fluxes may in fact support momentum deposition in the region of sharp density gradients*";

This position is further supported by the work of Dritschel and McIntyre (2008) and Dritschel and Scott (2011). In order to clarify this we have added these references and modified the text to make this point explicitly clear (L231-234): "*These internal gravity waves result in momentum transport to a critical layer near the velocity maximum, where wave breaking and momentum absorption locally accelerates the flow in a fashion analogous to the well-postulated models of the sharpening of zonal jets [cf. Dritschel and McIntyre, 2008; McIntyre, 2008; Buhler, 2014; Srinivasan and Young, 2012; Tobias et al., 2011; Baldwin et al., 2007; McIntyre and Palmer, 1983; Dunkerton et al., 2008; Wood and McIntyre, 2010; Dritschel and Scott, 2011]*."

Lines 261 et seq. & 276 and 277 You seem to be arguing that because the majority of gravity-current-generated bedforms are high aspect ratio, then the phenomenon you describe here should be the case generally in gravity currents. However, most of the turbidity current profiles measured to date in the ocean don't look like sharpened jets.

We respectfully disagree with this on four accounts: i) we argue the bedforms should be low-aspect ratio, as observed in field data (Symons et al., 2016) and in the present study; ii) hitherto no data set has existed for gravity current velocity and density profiles that includes their variation along longitudinal transects; iii) previous experimental and field studies have been predominately restricted to short duration flows, close to source, where internal-gravity-wave forcing is a result of a developing flow field; and iv) the data that does exist of other real-world turbidity currents, in steep canyon settings, does indeed show similar behaviour in the upper shear layer of the flow (the only component well resolved).

We have amended the text and added reference to Xu (2010), L240-242: "*Field measurements of turbidity currents in Monterey and Hueneme Canyon also show near-linear decay profiles of the upper shear layer [Xu, 10], although hitherto there has been little explanation of why this differs from the proposed Gaussian quadratic decay model [Kneller et al., 1999]*".

Fig 6a, lines 290-291. This seems a little simplistic. Even over a flat bed and with a stable upper boundary ($Rig > 0.25$) there will be large-scale coherent structures in the flow. Surely from what you say above, the waves are standing waves developed in response to the bed topography rather than as a consequence of upper boundary instabilities.

We agree, even over a flat-bed such coherent vortices will be shed into the flow. We have modified the text to note that boundary instabilities (including bedforms etc...) enhance the development of structure and internal-gravity-waves in turbulent flow, which now reads (L319-321): "*Smooth flow boundaries result in predominately parallel streamlines (a), with some internal variation due to turbulent fluid motion and coherent eddies shed into the flow,*

see Figure 1. Deformation of the lower and upper flow boundaries (b), e.g. by bedforms or the onset of Holmboe (Ho) or Kelvin-Helmholtz (KH) instabilities, enhances vertical movement within the flow."

If the velocity maximum acts like a free surface (l. 303) surely it will itself be prone to instabilities.

We agree that in an unstratified flow such an interface may be prone to instability. However, as a result of internal wave mixing of potential vorticity (i.e. conservation of circulation in a stratified environment) there will be an elasticity of velocity maximum which acts as a restoring force to the development of such instability. It is this gravity wave elasticity that generates the eddy transport barrier to mixing across the velocity maximum. We include reference to Dritschel and McIntyre (2008) to ensure this point is met.

Lines 310 – 312. If bedforms are the cause of this jet structure, why did Sequeiros et al. not observe them?

With reference to the response to lines 208-210, l. 211 and lines 261 276-277 above, these previous experimental and numerical data sets are unlikely to observe the evolution of the flow to an equilibrium forced by internal gravity waves. This is because of the short spatial length and time scales of such flume studies. Such experiments are incredibly useful but will have poor and limited representation of long run-out gravity current dynamics. However, as reflected in the amended summary section in the paper, the data we present will open new avenues for research, namely: i) to experimentally and numerically measure mixing dynamics in quasi-steady gravity currents and other stratified Earth surface flows; and ii) use conservation of circulation (i.e. Ertel's vorticity theorem) to investigate the temporal evolution of surface flows to internal wave forcing.

The amended text reads (L356-364): *"Several key questions arise from these new observations, including: i) the range of potential conditions supporting internal gravity wave development; ii) the importance of three-dimensional and statistically-unsteady flow processes; and iii) the time and length scales over which internal gravity waves affect mean flow dynamics in generic seafloor gravity currents. More generically the data presented is evidence for the need for further experimental and numerical quantification of mixing processes in stratified Earth surface flows and its quantification in terms of conservation of circulation (i.e. Ertel's vorticity theorem)."*

l. 328 Isn't Kneller et al., 2016 fairly relevant here?

Amended as requested.

Response to Reviewer #2

This paper demonstrates that the classical paradigm of oceanic gravity currents may not always apply, and indeed may be incorrect. The authors demonstrate, using unprecedented high quality measurements and analyses of the near sea-bed density and velocity fields, that gravity currents may rather behave like sharpened zonal jets in planetary atmospheres (and also like the Gulf Stream and other major ocean currents, though this is not noted in the paper). I believe this is a remarkable result of great insight, and I strongly support publication.

Noting the reviewer is one of the leading experts in the field of internal wave-turbulence interaction, the authors thank the reviewer for his extremely positive review of our paper. It is gratefully appreciated that this contribution was well received between traditional research disciplines. To address the reviewer's comments regarding the relationship to the Gulf Stream (and other oceanic currents), he is exactly right.

We mention this in the manuscript in Figure 4d and (L203-204): *"average velocity and density profiles in the Black Sea flow (Fig. 4c) in fact have more in common with those of oceanic and atmospheric zonal jets"*.

There is one issue that remains somewhat unclear; in lines 243-252 the authors correctly point out that the PV of a two-dimensional (x-z) flow is zero, and thus three-dimensional velocity and density variations are needed to explain the existence and sharpening of the PV in the gravity current. But I do not think this rules out this process occurring in a two-dimensional model of flow over a weakly varying bedform: density gradients may steepen by gravity-wave breaking and mixing, and the location of sharpest density gradients may naturally (on account of its greater stability) "attract" a more coherent flow less affected by turbulent mixing. In other words, flow along the sharp density gradient may not be as easily broken down as flow away from this gradient, and the eddy fluxes may in fact support momentum deposition in the region of sharp density gradients. This would

imply that the horizontal vorticity perpendicular to the plane of the assumed two-dimensional flow acts somewhat like PV in jet sharpening, even though this vorticity is subject to baroclinic sources (horizontal density variations). I believe this is a possibility and one that cannot yet be ruled out.

We agree and the reviewer is entirely correct that weak baroclinic sources may enable a quasi- two-dimensional flow to generate self-organisation if flow along the density gradient is negligible in comparison to momentum deposition in the region of sharp density gradients. The only caveat being that the weak baroclinic sources still result in a (weak) 3D flow.

The manuscript has been modified to acknowledge this and address the reviewer's point at lines L258-261: *"The exception to this being where weak lateral baroclinic sources enable self-organization of a quasi-two-dimensional flow; yet even here flow is three dimensional, with a lateral component along the density gradient."*

David Dritschel
University of St Andrews
6 September 2018

Response to Reviewer #3:

The paper is well written and very interesting. It described fascinating data on gravity currents from the Black Sea. The data is well described and is of significant importance to be published in Nature Communications.

We thank the reviewer for these very positive comments and the review of the research and data presented in our paper.

However, I do not agree with the theoretical explanation which has no mathematics to back it up. It argues by analogy primarily with planetary scale phenomena. I believe that most of the speculation should be removed and that the paper would then be suitable for publication. Better yet would be a theoretical calculation, like in the McIntyre, that actually shows that their proposed mechanism is correct, though since I think it is wrong I doubt this is possible.

We have seemingly confused the reviewer in the way we have presented these arguments. We believe that confusion has arisen between the comparison of zonal jets and the gravity currents presented. The concept of PV conservation is not exclusive to flows in a rotating framework. PV is merely the sum of the inherent vorticity of a flow and any additional component from a rotating framework and extend to any flows modified by dispersive waves. As evidenced by the comments of the second reviewer (Dritschel), the theory presented is not speculation but is a translation of existing theory that has been applied to explain self-organisation in flows in both rotating and non-rotating reference frames (cf. Dritschel and McIntyre, 2008). We have amended the text to flag that this is the central reference (L65, 68) and we also discuss examples of conservation of vorticity in a non-rotating framework.

We have also modified the text to reflect this point, at L63-64: *"Planetary Rossby waves, and other types of dispersive waves [Dritschel and Scott, 2008], where phase speed varies with wavelength, can generate systematic correlations of turbulence (radiation stresses) and enable up-gradient momentum transport [Buhler, 2014]."*

And L94-95: *"Self-organisation of turbulent flows by dispersive waves is not limited to the formation and self-sharpening of zonal jets by planetary Rossby waves, nor to flows in rotational frames of reference."*

We have also modified the paper summary, which now reads L356-364: *"Several key questions arise from these new observations, including: i) the range of potential conditions supporting internal gravity wave development; ii) the importance of three-dimensional and statistically-unsteady flow processes; and iii) the time and length scales over which internal gravity waves affect mean flow dynamics in generic seafloor gravity currents. More generically the data presented is evidence for the need for further experimental and numerical quantification of mixing processes in stratified earth surface flows and its quantification in terms of conservation of circulation (i.e. Ertel's vorticity theorem)."*

We anticipate that these changes will address all the reviewer's concerns over the speculation they thought was present in the paper.

The use of the word "dissipation" instead of "diffusion" does not add to the confidence of the reader in the theoretical understanding of the authors.

We thank the reviewer for highlighting this unfortunate and embarrassing typographic error. Indeed the classical model is that short-scale turbulence "diffuses" flow properties. However, wave-turbulence interaction results in long-range radiative stresses that deposit momentum (cf. reviewer 2, comment 2) where wave-breaking in regions of sharp velocity, as a retrograde frictional force, results in the dissipation of wave energy (Dritschel & McIntyre, 2008). We have changed the text accordingly.

I've read reference (25) McIntyre, M. E. (2008). "Potential-vorticity inversion and the wave-turbulence jigsaw: some recent clarifications. *Advances in Geosciences*, 15, 47-56." And I have trouble seeing how it is an explanation for possible sharpening in gravity currents. It is concerned with large scale rotating flows where the key idea is the invertibility of potential vorticity, that is at any time the potential vorticity hold all the information about the flow and the velocity fields and density fields can be calculated from it. Rotation is only marginally important for only the largest gravity currents and the "invertibility" of potential vorticity does not hold....for example the McIntyre says "A full understanding of such phenomena can scarcely be arrived at without the conscious or unconscious use of the PV "invertibility principle" or something equivalent (e.g., Hoskins et al., 1985 & refs). It is an essential component of the conceptual "Lego set" one needs."

As discussed, clarified and addressed in the above responses and the subsequent changes we have made to the manuscript, the rotational framework of the flow is not the key idea behind potential vorticity invertibility. The conceptual "Lego Set" is built on conservation of circulation in a stratified environment (a simplification of Ertel's vorticity theorem). This is equally applicable in rotating and non-rotating frames of reference. We think the changes we have made above will mean that the points and arguments we are making in this regard are now clear.

I do not see therefore how this paper can explain the phenomena discussed in this paper.

We refer to the previous responses, and amendments to the text, detailed above, which have now hopefully clarified the use of the theory to explain the observations. The authors also refer to the second reviewer (Dritschel) who states: "*The authors demonstrate, using unprecedented high quality measurements and analyses of the near sea-bed density and velocity fields, that gravity currents may rather behave like sharpened zonal jets in planetary atmospheres [...]. I believe this is a remarkable result of great insight, and I strongly support publication.*"

line 18 "isotropic turbulent fluctuations" The turbulence is non-isotropic in regions influenced by the boundary

We agree that the turbulence is anisotropic near the boundary (and indeed likely throughout the flow). However, extant theoretical models of turbulent diffusion are based on the concepts of isotropic turbulent diffusion (Kraichan, 1987). We have made no changes in response to this comment.

Figure 1 The figure says "dissipation of momentum" but momentum is conserved; It cannot be "dissipated". In this and many other places (i.e line 66) in the manuscript, dissipated should be changed to diffusion.

Amended as requested.

Line 71 "the self-organisation of turbulence, by dispersive internal waves, results in up-velocity-gradient momentum transport" This needs a reference. I find this explanation quite hard to believe

Coherent structures (e.g. dispersive waves and coherent vortices) readily result in anisotropic turbulent fluctuations and thus momentum and material transport. For clarity we have added reference to Dritschel and McIntyre (2008) in the figure caption for the interested reader.

Line 72 "whilst irreversible wave breaking transfers wave energy into the mean flow." This also doesn't seem right to me. wave breaking will increase the turbulence intensity but won't increase the mean flow.

We agree that this is counter-intuitive. However, this concept has been well established in the atmospheric sciences, where shear provides a retrograde force resulting in dispersive wave breaking. The change in angular momentum distribution that result from wave-breaking has been shown to result in jet-sharpening. We refer to the second reviewer (Dritschel) "*density gradients may steepen by gravity-wave breaking and mixing, and the*

location of sharpest density gradients may naturally (on account of its greater stability) "attract" a more coherent flow less affected by turbulent mixing. In other words, flow along the sharp density gradient may not be as easily broken down as flow away from this gradient, and the eddy fluxes may in fact support momentum deposition in the region of sharp density gradients".

In response and to ensure clarity of the argument, we have in our revision modified the text to reflect this point (L68-70): "*Irreversible breaking of Rossby, or other, dispersive waves, near a critical layer where the background flow speed tends to the wave phase speed, results in deposition of wave momentum, with concomitant changes in angular momentum distribution [Dritschel and McIntyre, 2008], and generation of mean flow [Srinivasan and Young, 2012; Tobias et al., 2011].*"

line 80 "negative viscosity" I would say that this is highly controversial and any model that includes things in this form is highly phenomenological and technically speaking is ill-posed and admits no grid independent solutions.

Negative viscosity was the historical term (cf. Starr, 1968) used to describe what is now understood as anti-frictional radiation stresses that are the result of "*dynamical organization of fluctuations*" (Dritschel and McIntyre, 2008; McIntyre, 2003, *Stellar Astrophysical Fluid Dynamics*, 111-130). Essentially this term implies that standard Reynolds stress models are poorly suited to describe non-homogenous stratified fluids (cf. Yeh et al. (2013) Turbidity current with a roof: Success and failure of RANS modeling for turbidity currents under strongly stratified conditions. *Journal of Geophysical Research: Earth Surface*, 118, 1975-1998) and associated wave-turbulence interaction. Whilst we understand the reviewer's unease, due to common usage in the earth science community and the long use of this terminology, we argue it is valuable to leave it in, but have added "*anti-frictional radiation stresses*" (L60) to the text to be clear to the reader the meaning of the term.

Reviewer #1 (Remarks to the Author):

In general the authors have addressed my doubts well. There remain a couple of things.

I don't think that Nielsen & Teakle "established that turbulent diffusion is fundamentally flawed when turbulent motion results in mixing at scales larger than flow variable length scales". According to my reading, they demonstrate the effectiveness of Fickian diffusion in describing the suspended sediment distribution in an Eulerian reference frame. Do they actually mean Absi's (simulation-based) comment on this? Surely diffusion describes rather well the time-averaged distribution of sediment in natural and experimental turbidity currents and in fully-resolved (DNS) simulations. As I said before, this is a little out of my field, so others may have better-informed comment.

I am still not entirely convinced of the comparison with turbidity currents. As far as the comments on the shape of the velocity profile is concerned, the Gaussian profile probably only applies to Froude subcritical flows (if that means anything) whereas the Monterey and Hueneme flows reported by Xu would have been supercritical. The comparison with the Congo flows is spurious in the context in which they make it, in that the flows that Azpiroz-Zabala et al describe are not 'typical turbidity currents'. However, ironically, these flows (as well as those recently described by Paull et al from the Monterey Canyon) may actually be more similar to what the present authors describe from the Black Sea in that there may be a sharper interface within them, perhaps at a velocity maximum.

I've made a few additional comments on the ms, but these are the main points.

I recommend publication subject to addressing these points

Reviewer #3 (Remarks to the Author):

The authors have replied in detail and satisfactorily to my comments and I recommend that the paper to be published as is.

Response to Reviewers Comments

We would like to thank the reviewers for their detailed and thoughtful commentaries on our manuscript. We have addressed (as highlighted in blue below) all the comments of each of the reviewers and in our response, below, we detail the action we have taken in relation to each point. The line numbers refer to location of the changes we have made in the new (revised) manuscript, where we have tracked all the changes to allow full transparency in order to aid the editing process.

Reviewer #1 (Remarks to the Author):

In general the authors have addressed my doubts well. There remain a couple of things.

1) I don't think that Nielsen & Teakle "established that turbulent diffusion is fundamentally flawed when turbulent motion results in mixing at scales larger than flow variable length scales". According to my reading, they demonstrate the effectiveness of Fickian diffusion in describing the suspended sediment distribution in an Eulerian reference frame.

We firstly note that this is a comment on our response to the first set of reviews and that the Nielsen and Teakle (2004; *Physics of Fluids*, 16, 2342) reference referred to here is not directly cited in the manuscript.

In Reviewer #1's original comments they asked if, in general, the [Fickian] "turbulent diffusion of sediment away from the bed is a necessary process for the maintenance of the suspension".

As part of a broader response to this comment, the work of Nielsen & Teakle (2004) was advanced in the revised manuscript, and is clarified further below. In Nielsen and Teakle (2004) the first order Prandtl mixing model is shown to result in the standard Fickian turbulent diffusion model. However, Nielsen and Teakle (2004) demonstrate that the 2nd order term in the expansion of the Prandtl mixing model depends on the third derivative of the sediment concentration. Nielsen and Teakle (2004) go on to state explicitly that "*Thus, the variability of the magnitude of the bracketed function [the sum of first and second derivative] in the expressions for e_{Fick} [apparent Fickian diffusivity] and v_t [eddy viscosity], gives the extent to which Reynolds hypothesis 'Same Fickian diffusivity for momentum and particles of different sizes' is violated*".

Analysis of the bracketed function describing turbulent mixing yields a key result: that the ratio of mixing to the first order gradient is not forced to be strictly positive. Therefore, depending on the higher order gradients, and thus the length-scales over which mixing occurs, the effective viscosity is not even forced to be strictly positive as prescribed, *a priori*, as in the standard Fickian eddy diffusivity model. Thus, models relying on the universality of these assumptions, without resolving large-scale flow mixing, are potentially flawed. Therefore, we agree with the reviewer that our summary statement of '*fundamentally flawed*' was poorly chosen, and it would have been more appropriate to state that the "same Fickian diffusivity for momentum and particles of different sizes may be violated", and, therefore, models based on these assumptions are potentially flawed. We have thus made this change.

Whilst the manuscript does not set out to propose a new turbulent mixing closure, the authors believe the internal-gravity-wave driven large-scale mixing model, which has been shown to drive long-range momentum and mass transport (cf Dritschel & McIntyre, 2008), thus has significant implications on the suspension of sediment in turbulent stratified flows.

In order to address the reviewers comments, and to clarify this point on turbulent mixing processes, in the main manuscript text, we have amended Lines 338-340 to read:

"More generically the data presented are evidence for the need for further experimental and numerical quantification of turbulent mixing processes in sediment-laden, stratified Earth

surface flows and its quantification in terms of conservation of circulation (i.e. Ertel's vorticity theorem)."

2) Do they actually mean Absi's (simulation-based) comment on this [Nielsen & Teakle, 2004]? Surely diffusion describes rather well the time-averaged distribution of sediment in natural and experimental turbidity currents and in fully-resolved (DNS) simulations. As I said before, this is a little out of my field, so others may have better-informed comment.

We again note that this is a comment on our response and actions to the first set of reviews.

We agree with the reviewer that the first order, diffusion based, Fickian model can describe sediment suspension well, where turbulent mixing occurs at small-scale. However, even these models depend on correct parameterization of a quasi-empirical mixing length scale (e.g. Absi, 2005 (Physics of Fluids, 17, 079101), 2011 (The Proceedings of the Coastal Sediments, Miami, Florida, USA, 2 May – 6 May 2011, edited by Julie D Rosati, Ping Wang & Tiffany M Roberts, World Scientific Publishing, pages 1096 – 1108) that the reviewer refers to. Where mixing is at small-scale and mixing-length resolved - as in unstratified, unforced or under-developed turbulent flow, which comprise the majority of experimental and DNS studies to date - it is thus unexpected that diffusion models perform well. However, turbulent mixing in stratified flows results in the development of dispersive internal-gravity-waves. We argue in the manuscript that dispersive-waves have already been demonstrated to result in the organization of dynamical fluctuations and thus long-range momentum (and thus material) transport, not described by Fickian diffusion (i.e. radiation stresses, cf Dritschel & Scott, 2008 and previous comments of reviewer 2). The role of dispersive internal waves is highlighted on Lines 64-66,

"Planetary Rossby waves, and other types of dispersive waves [Dritschel & McIntyre, 2008], where phase speed varies with wavelength, can generate systematic correlations of turbulence (radiation stresses) and enable up-gradient momentum transport [Buhler, 2014]."

As the role of dispersive internal waves are unlikely to have been identified in previous experimental and numerical simulations of gravity currents because the time-scales are too short and the degree of stratification limited, we further note, on Lines 280-281 that

"These new observations require evaluation of the time and length scales over which internal-gravity-wave forced flows converge to a pseudo-steady state."

We hope that we have provided improved clarity for the reviewer. However, given the broader arguments are described in the main manuscript, and the wide audience that the paper is aimed at, we feel that the addition of more detailed discussion is not appropriate for the main manuscript. We would of course be willing to add this in should the editor wish, but on balance and noting that the reviewer admits this is beyond their field of study, we would prefer not to confuse the flow of the paper with such detail. We also note that as reviews will be made available as supplementary material online, then this detail is available for interested parties.

3) I am still not entirely convinced of the comparison with turbidity currents. As far as the comments on the shape of the velocity profile is concerned, the Gaussian profile probably only applies to Froude subcritical flows (if that means anything) whereas the Monterey and Hueneme flows reported by Xu would have been supercritical.

The relationship between Gaussian profiles and Froude number is an argument made by Kneller et al. (2016), where it is argued that Gaussian profiles likely only relate to subcritical gravity currents and that supercritical currents may be close to linear or exponential. In contrast, however, Sequeiros et al. (2010), in their extensive suite of experiments, note for supercritical currents that *"a combination of logarithmic and Gaussian profiles could also be used"* [to fit to the profiles]. This is further supported by observations and conclusions of Altinakar, Graf and Hopfinger (1996) [Flow structure in turbidity currents, Journal of Hydraulic Research, 34(5), 713-718]. There is therefore an interesting argument on the role

to which flow supercriticality determines the degree to which the upper shear layer reflects a Gaussian profile, but this is not one where there has been extensive analysis of the differences between subcritical and supercritical currents. However, to better reflect this discussion and address the reviewers point in this regard, we have modified the introductory text, Line 44-46 to state:

"with an exponentially decreasing (concave up) profile, with a Gaussian decay with distance from the velocity maximum, in subcritical [Kneller et al., 1999; Sequerios et al., 2010; Kneller et al., 2016] and supercritical flows [Sequerios et al., 2010]; albeit it has been argued that linear or exponential decay may occur in some supercritical currents [Kneller et al., 2016]"

We also note that the Black Sea data are unique for oceanic gravity currents as full velocity and directly-sampled density profiles were resolved in detail, demonstrating that the data analysed was of a subcritical flow (where as the reviewer notes, and we now make clearer, the Gaussian model should best apply). We have therefore modified the paper accordingly, on Lines 135-138:

"The flow is highly turbulent, with a Reynolds number $Re \sim 2.5$ million, and subcritical, with a bulk Froude number $Fr \sim 0.6$ (Fig. 3), calculated from directly-sampled velocity and density data."

Furthermore, we have then made the subcritical nature of the flow explicitly clear when discussing the nature of the profile in the upper shear layer of the Black Sea flows, as identified in the revised text, Line 142-144:

"secondly the linearly exponentially-decreasing, rather than Gaussian, form of the upper shear layer (as highlighted by the rapid decrease in flow velocity just above the velocity maximum, see insets in Fig. 3b) in this subcritical flow."

We agree with the reviewer that the field data from Monterey, Congo or Hueneme have been interpreted as supercritical - albeit they have no direct density data (although inference has been made from acoustic backscatter) and thus Froude numbers are not fully calculated. However, we have rewritten the text to recognise that these linear zones have previously been interpreted as evidence for supercritical flow, whilst also indicating that an alternative assessment is that these are the product of self-organization via internal-gravity-wave forcing, Line 193-196:

"Measurements of turbidity current velocity in Monterey and Hueneme canyons also show near-linear decay profiles of the upper shear layer [Xu, 2010], previously interpreted as a product of supercritical flow [Kneller et al., 2016] (see earlier discussion). Alternatively, these profiles are plausibly explained by self-organization via internal-gravity-wave forcing [Dritschel and McIntyre, 2008]."

4) The comparison with the Congo flows is spurious in the context in which they make it, in that the flows that Azpiroz-Zabala et al. describe are not 'typical turbidity currents'. However, ironically, these flows (as well as those recently described by Paull et al. from the Monterey Canyon) may actually be more similar to what the present authors describe from the Black Sea in that there may be a sharper interface within them, perhaps at a velocity maximum.

We have removed reference to the Azpiroz-Zabala et al. (2017) paper in terms of using this as an example for the height of the velocity maximum (as requested in the referee's comments on our response to the reviewers), and replaced it with a more appropriate paper (see response to later point below). The Azpiroz-Zabala et al. (2017) paper is now solely used elsewhere as one of a number of examples of turbidity current flows for comparison.

Moreover, we agree with the reviewer that, if driven by a high-concentration lower layer, gravity current dynamics may not be directly relevant to the analysis proposed of low-concentration flows detailed in our manuscript. The data from Paull et al. (2019) details acoustic backscatter of an unresolved, but likely high concentration near bed layer $\gg 10\%$ concentration v/v , in a short-lived near-to-source gravity current. In contrast Azpiroz-Zabala

et al. (2017) report multiple flows, with peak near-bed concentrations estimated at <3% v/v at the flow head. In the body of the flow estimated flow concentration is even more dilute, ~0.1% v/v. Further, Azpiroz-Zabala et al. (2017) report that the turbidity current is characterized by fine-grained muddy sediment, which has negligible settling velocity. Saline gravity currents have long been used as an analogue for such dilute fine-grained sediment-laden turbidity currents (cf Altinakar, Graf & Hopfinger, 1996; Sequerios et al. 2010). As such we argue that the Azpiroz-Zabala et al. (2017) paper is appropriate for comparison as one of a number of examples, albeit we agree with the referee that the specific comparison we had made in terms of the height of the velocity maximum in turbidity currents is much better addressed with a different reference. We have modified the text accordingly, and provide details of this in response to the additional comments on the manuscript, below.

I've made a few additional comments on the ms, but these are the main points.

Below we address the attached comments on the response to review in the order they occur.

- “I don't think this is what Nielsen & Teakle say at all.”

We expand on our arguments on the shortcomings of an eddy diffusivity model in points 1) and 2) above, concluding that the higher order terms of the Prandtl mixing model presented by Nielsen and Teakle (2004) may result in non-diffusive mixing. We have made this clearer in the changes we have made (detailed above).

- “the fact of linearity does not of itself imply a particular mechanism, does it?”

The reviewer is correct in that correlation does not imply causation. We have modified the text on Line 193-196 to read:

“Measurements of turbidity current velocity in Monterey and Hueneme canyons also show near-linear decay profiles of the upper shear layer [Xu, 2010], previously interpreted as a product of supercritical flow [Kneller et al., 2016] (see earlier discussion). Alternatively, these profiles are plausibly explained by self-organization via internal-gravity-wave forcing [Dritschel and McIntyre, 2008].”

- “The Monterey Canyon flows are supercritical (if that means anything), to which the Gaussian model would not apply (if, indeed, it applies to anything).”

We have addressed this point as discussed above and we have modified the text accordingly on Line 44-46 to state:

“with an exponentially decreasing (concave up) profile, with a Gaussian decay with distance from the velocity maximum, in subcritical [Kneller et al., 1999; Sequerios et al., 2010; Kneller et al., 2016] and supercritical flows [Sequerios et al., 2010]; albeit it has been argued that linear or exponential decay may occur in some supercritical currents [Kneller et al., 2016]”

- “This is not the appropriate reference here; something about 50 years earlier might be more appropriate - especially as the particular currents that Azpiroz-Zabala et al. describe are far from analogous with saline gravity currents, having a rapidly moving and exceptionally dense layer at the base, whose nature is unclear, and which may be driving the overlying current. It may, in fact, be representative of the process you describe, but that is not the context in which you seem to be using the citation here.”

We agree and have replaced this reference with Peakall and Sumner (2015) who synthesise the data on the height of the maximum velocity. The reviewer is correct that this goes back to at least the work of Tesaker (1969), 50 years ago. However, we feel that this recent synthesis is more useful, and more accessible than Tesaker's 1969 PhD thesis. In terms of other comparisons, we agree with the reviewer on the fact that this example has some differences to a classical turbidity currents, as discussed in point 4 above. We could remove reference to this paper entirely if really needed, however we feel that this is a high profile paper, with the most detailed dataset from a field scale turbidity current, with many commonalities with our analysis, and we thus argue it is worth incorporating as one amongst several examples we use to highlight the broader importance of our findings.

- “see comment above about criticality; I think that is the explanation, of which ‘*hitherto there has been little*’, to use your own words.”

We acknowledge that previous work has highlighted the potential role of flow supercriticality in determining gravity current velocity profiles. As discussed in response to point (3) above, we have amended the manuscript to state on Line 44-46:

"with an exponentially decreasing (concave up) profile, with a Gaussian decay with distance from the velocity maximum, in subcritical [Kneller et al., 1999; Sequerios et al., 2010; Kneller et al., 2016] and supercritical flows [Sequerios et al., 2010]; albeit it has been argued that linear or exponential decay may occur in some supercritical currents [Kneller et al., 2016]"

and Line 193-196 to read:

"Measurements of turbidity current velocity in Monterey and Hueneme canyons also show near-linear decay profiles of the upper shear layer [Xu, 2010], previously interpreted as a product of supercritical flow [Kneller et al., 2016] (see earlier discussion). Alternatively, these profiles are plausibly explained by self-organization via internal-gravity-wave forcing [Dritschel and McIntyre, 2008]."

- “Are we using high and low aspect ratio in the same sense? Symons et al.'s compilation shows typical values of 100:1”

No, we have been using high and low aspect ratio differently. We apologise for the confusion and thank the reviewer for highlighting this. We have thus modified the text to state ‘high aspect’ rather than ‘low aspect’ to map to that of Symons. Our results are higher than those in Symons et al. 2016 who report values up to aspect ratios of ~150:1, albeit this is based on limited data at the top end. We have changed the text on Lines 114-115:

"channel floor is ornamented by high aspect ratio, ~200:1, sedimentary bedforms (Fig. 2b)"
and on lines 227-228,

"Despite normally being much larger than their terrestrial analogues, seafloor gravity currents typically form long wavelength, low amplitude high aspect-ratio bedforms [Symons et al., 2016]."

and on lines 240-242,

"That bedform evolution is self-limited by the dynamics of the near-bed velocity maximum, not the depth of the entire flow, may therefore explain the enigmatic high aspect-ratio of bedforms sculpted by seafloor gravity currents."

I recommend publication subject to addressing these points

We thank the reviewer for their comments which have greatly improved the clarity of the manuscript, and for their positive response to the work presented.

Reviewer #3 (Remarks to the Author):

The authors have replied in detail and satisfactorily to my comments and I recommend that the paper to be published as is.

We thank the reviewer for their review of the research and data presented in our paper and for their positive response. We are glad that our reply fully addressed their original comments.